# Radiosensitization-Related Cuproptosis LncRNA Signature in Non-Small Cell Lung Cancer

**DOI:** 10.3390/genes13112080

**Published:** 2022-11-09

**Authors:** Qiushi Xu, Tong Liu, Junjie Wang

**Affiliations:** 1Institute of Medical Technology, Peking University Health Science Center, 38 Xueyuan Road, Haidian District, Beijing 100010, China; 2Department of Radiation Oncology, Peking University Third Hospital, 49 Huayuan North Road, Haidian District, Beijing 100191, China; 3Center of Basic Medical Research, Institute of Medical Innovation and Research, Peking University Third Hospital, 49 Huayuan North Road, Haidian District, Beijing 100191, China

**Keywords:** radiotherapy, cuproptosis, lncrna, non-small cell lung cancer, immunity

## Abstract

A new treatment modality targeting cuproptosis is gradually entering the public horizon. Cuproptosis is a new form of regulated cell death distinct from ferroptosis, apoptosis, autophagy, and necrosis. Previous studies have discovered that the copper level varies considerably in various cancers and that an increase in copper content is directly associated with the proliferation and metastasis of cancer cells. In non-small cell lung cancer (NSCLC) after radiation, the potential utility of cuproptosis-related long noncoding RNAs (lncRNAs) is still unclear. This research aimed to develop a prediction signature based on lncRNAs associated with cuproptosis to predict the prognosis of NSCLC patients following radiation. Methods: Expression data of primary tumors and adjacent solid tissues were downloaded from The Cancer Genome Atlas (TCGA) database, along with the corresponding clinical and mutational data. Univariate and multivariate COX analyses and LASSO regression analyses were performed to obtain a predictive signature of lncRNAs associated with cuproptosis. The data were randomly grouped into a training group used for model construction and a test group used for model validation. The model was validated by drawing a survival curve, risk curve, independent prognostic analysis, ROC curve PFS analysis, etc. Results: The lncRNA signature consisting of six cuproptosis-related lncRNAs (AC104088.1, PPP4R3B-DT, AC006042.3, LUCAT1, HHLA3-AS1, and LINC02029) was used to predict the prognosis of patients. Among them, there were three high-risk lncRNAs (LUCAT1, HHLA3-AS1, and LINC02029) with HR > 1 and three protective lncRNAs (AC104088.1, PPP4R3B-DT, and AC006042.3), with an HR < 1. Data analysis demonstrated that the cuproptosis-related lncRNA signatures could well predict the prognosis of NSCLC patients after radiation. Patients in the high-risk category receive a worse prognosis than those in the low-risk group. Cuproptosis-related risk prediction demonstrated better predictive qualities than age, gender, and pathological stage factors. Conclusion: The risk proposed model can independently predict the prognosis of NSCLC patients after radiotherapy, provide a foundation for the role of cuproptosis-related lncRNAs in NSCLC after radiotherapy, and provide a clinical strategy for radiotherapy combined with cuproptosis in NSCLC patients.

## 1. Introduction

Lung cancer is the second most prevalent malignant tumor in the world, with the greatest fatality rate [1]. Radiation therapy (RT) is a vital therapeutic strategy for lung cancer. Previous research showed that 77% of the patients with lung cancer have evidence-based criteria for radiation treatment [2]. Additionally, radiation improved the 5-year local control rate by 8.3% and survival rate by approximately 4% [3] and has been utilized as a decisive treatment for early inoperable tumors [4] and locally progressive disease [5,6] in non-small cell lung cancer (NSCLC). The use of radiation treatment has improved as NSCLC patient survival rates have increased. Technology has permitted radiation therapy to target tumors more precisely and lessen inadvertent irradiation of nearby normal tissue. This has increased the use of radiation for lung cancer; nevertheless, further research is required to raise survival rates and lower toxicity. It is crucial to identify non-small cell predictive indicators to support novel treatment targets.

In 2022, a new type of cell death, Cuproptosis, was proposed for the first time [7]. Cu ions are involved in biological engineerings, such as cellular respiration and antioxidant reduction systems [8,9]. Cells that rely on mitochondrial respiration have a high sensitivity to copper-induced cell death and have been effectively validated in lung cancer cells [10]. However, the mechanism underlying its occurrence and development requires further study. Prior studies have shown that elevated serum copper levels inhibit the response to radiotherapy [11], while copper chelators can improve their effectiveness [12]. These findings suggest that targeting copper-related lncRNAs may provide a new strategy for radiosensitization.

Long noncoding RNAs (lncRNAs) have become essential players in the development, progression, and management of a variety of solid tumors and hematological malignancies [13,14,15]. The lncRNA *HOX* transcript antisense RNA (HOTAIR) increases lung cancer cell proliferation, survival, invasion, metastasis, and treatment resistance by enlisting chromatin modifiers to inhibit gene expression [16]. LncRNA CARLo-5 regulates cell cycle, proliferation, and invasion and is upregulated in NSCLC with poor prognosis [17]. By altering the expression of MMP-2/-9, LncRNAs linked with microvascular invasion in hepatocellular carcinoma (MIVH) control the growth and invasion of squamous cell carcinoma. There is currently limited research on cuproptosis-related lncRNAs and none on cuproptosis-related lncRNAs in NSCLC after radiation.

In this research, we developed a prognostic model based on cuproptosis-related lncRNAs, evaluated their utility in determining prognosis and tumor immune infiltration in NSCLC patients after radiation, and carried out internal validation.

## 2. Materials and Methods

### 2.1. Data Sources

The dataset and clinical follow-up information after radiotherapy were downloaded from (https://portal.gdc.cancer.gov/, accessed on 10 June 2022). We collected the data from 180 primary tumors, 18 normal solid tissue samples, and the clinical data of 178 patients who underwent radiation therapy. Furthermore, the data on the cuproptosis-related genes were collected through a literature search [10,18,19,20].

The inclusion criteria: (1) bronchi and lungs were screened in the database; (2) squamous cell carcinoma and adenocarcinoma were screened by disease type; (3) primary tumor and solid normal tissue were screened by sample type, and (4) transcriptome analysis was performed.

The exclusion criterion was incomplete data.

#### 2.1.1. Construction of Predictive Signatures with Cuproptosis-Related LncRNAs

The “limma” package was used to calculate the correlation between the cuproptosis-related genes and lncRNAs. corFilter = 0.4 and *p*-value Filter < 0.001 as the screening criteria, 535 lncRNAs were identified (Appendix A). Differential lncRNAs were identified using univariate Cox regression analysis to identify the lncRNAs associated with the prognosis of patients with NSCLC, and then multivariate Cox analysis was performed on the selected genes. A stepwise regression method (stepAIC) further reduced the number of genetic risk factors. Subsequently, a predictive model was built with the following formula:Risk Score =  x1 ∗ coef1 x2 ∗ coef2 … xn ∗ coefn

Here, the coefficient (coef) value and x represent the lncRNA expression value for model construction.

#### 2.1.2. Survival Analysis and PFS Analysis

TGCA data were divided into training and validation sets. The training and validation datasets were matched 1 to 1. “survival” and “survminer” package were used to visualize different groupings, and pan-cancer data were downloaded from https://xenabrowser.net, accessed on 31 October 2022.

#### 2.1.3. Co-Expression Network

The “limma” package is used to analyze co-expression relationships. Using the correlation coefficient corFilter = 0.4 and *p*-value Filter = 0.001 as the filter criteria, the Cytoscape software was used to analyze the model construction.

#### 2.1.4. Nomo and Calibration Plots

Combining the risk scores with clinical information, nomogram survival models for OS were built using the R package “rms” based on univariate and multivariate outcomes. The calibration curve estimate was adjusted for optimism by applying a bootstrap procedure (B = 1000 repetitions).

#### 2.1.5. Tumor Immunoassay

The immune infiltrating cell file download address (http://timer.comp-genomics.org) was accessed on 18 June 2022. The immune escape file download address (http://tide.dfci.harvard.edu/) was accessed on 23 June 2022, using the “limma” and “pheatmap” packages.

#### 2.1.6. Tumor Mutational Burden Analysis

Mutation data for NSCLC after radiotherapy were downloaded from the TCGA database. The 50 genes with the highest mutation frequencies were analyzed using the “maftools” package. The tumor mutational burden was subjected to survival analysis using the “maftools” package.

#### 2.1.7. Culture of Cells

At 37 °C and 5% CO_2_, A549 and H1299 cell lines were incubated in DMEM media (Gibco, Waltham, MA, USA) with 10% fetal bovine serum (FBS; Gemini, New York, NY, USA) and 10% streptomycin-penicillin (Gibco).

### 2.2. Quantitative Real-Time PCR

TRIzol reagent was used to extract total RNA from radiotherapy-treated A549 and H1299 cells for quantitative real-time PCR (Invitrogen, Carlsbad, CA, USA). HiScript III RT SuperMix for qPCR (+gDNA wiper) was used to convert the RNA into cDNA (Vazyme Biotech Co., Ltd., Nanjing, China). To determine the expression level of lncRNA-LINC0209, real-time PCR was performed using SuperReal PreMix Plus (SYBR Green; Tiangen Biotech, Beijing, China) and a QuantStudio5 real-time PCR apparatus (Applied Biosystems, Foster City, CA, USA). Relative gene levels were calculated using the 2^−ΔΔCT^. Table 1 shows the sequences of the primers used in PCR.

### 2.3. Statistical Analysis

All statistical analyses were performed using the R software (China, https://mirrors.tuna.tsinghua.edu.cn/CRAN/, accessed on 23 June 2022, version 4.1.3).

## 3. Results

### 3.1. Construction of the Cuproptosis-Related LncRNA Predictive Signature

Our overall research flowchart is shown in Figure 1. The limma software package visualized the relationship between the co-expressions of lncRNA and cuproptosis-related genes in the 198 samples. The co-expression relationship is shown through a Sankey diagram; the data are presented in Appendix A and Figure 2A. The correlations between the cuproptosis-related genes and lncRNAs were further determined using correlation coefficients with the cutoff limits of corFilter = 0.4 and *p*-value Filter < 0.001 as the screening criteria. A total of 535 cuproptosis-related lncRNAs were identified using univariate COX analysis. Multivariate Cox regression analysis revealed six lncRNAs associated with cuproptosis (AC104088.1, PPP4R3B-DT, AC006042.3, LUCAT1, HHLA3-AS1, and LINC02029), which were used to construct the correlation prediction model (Appendix A). Risk score = (−0.760406773663811 × AC104088.1 expression value) + (−0.719677929959793 × ‘PPP4R3B-DT’ expression value) + (−2.86161382387809 × AC006042.3 expression value) + (0.3524580171184 × LUCAT1 expression value) + (0.935357156077803 × ‘HHLA3-AS1’ expression value) + (0.918845627145671 × LINC02029 expression value). Among these, there were three high-risk genes (LUCAT1, HHLA3-AS1, and *LINC02029*) with HR > 1, and three protective genes (AC104088.1, PPP4R3B-DT, and AC006042.3) with HR < 1. The hazard ratios are shown in Figure 2B. The heat map showed that *NLRP3, MTF1, LIPT1, GLS,* and *ATP7A* were positively regulated by lncRNA-AC006042.3, while *PDHA1, NFE2L2*, and *LIPT2* were negatively regulated by lncRNA-AC006042.3. The *PDHA1, NFE2L2, LIPT2, GCSH, DLST,* and *DLD* genes were positively regulated by lncRNA-AC104088.1. *NLRP3* and *GLS* genes were negatively regulated by lncRNA-AC104088.1. *NLRP3*, *MTF1*, *GLS*, and *ATP7B* genes were positively regulated by lncRNA-HHLA3-AS1, while the *GLS* and *ATP7A* genes were positively regulated by lncRNA-LINC02029. *LIPT1*, *GLS*, and *ATP7A* were positive regulators of lncRNA-LUCAT1. *PDHA1*, *LIPT2*, *LIPT1*, and *CDKN2A* genes were positively regulated by lncRNA-PPP4R3B-DT. The *NLRP3*, *MTF1*, and *ATP7A* genes negatively regulate lncRNA-PPP4R3B-DT (Figure 2C).

Next, we used the Cytoscape software to further analyze the co-expression relationship between the cuproptosis-related genes and lncRNAs. The results showed that *GLS* was co-expressed with AC006042.3, LUCAT1, and LINC02029, *ATP7A* was co-expressed with HHLA3-AS1, *CDKN2A* was co-expressed with PPP4R3B-DT, and *NFE2L2* was co-expressed with AC104088.1 (Figure 2D). We further verified the reliability of the above results obtained from the TCGA database and selected LINC02029 with the highest risk factor for qPCR correlation verification in cell lines. We irradiated cells with X-rays at a dose rate of 6 Gy/min for 0 or 8 Gy and collected RNA for qPCR 48 h after radiotherapy. LINC02029 in A549 and H1299 (Figure 2E) NSCLC cell lines after radiotherapy have higher expression than without radiotherapy.

### 3.2. Validation of Predictive Signature and Prognosis

We randomized 177 NSCLC patients (one normal sample was removed) who underwent radiotherapy into two groups (*n*1 = 88, *n*2 = 99) to verify the applicability of OS prediction features based on the entire TCGA dataset. The demographic characteristics of the patients in both cohorts are shown in Table 2.

The risk score of each patient was calculated according to the formula, and the median risk score was used to divide the patients into high-risk and low-risk groups. To determine the value of the risk score in predicting the prognosis of patients with NSCLC after radiotherapy, library (survival) and library (survminer) analyses were performed to analyze the overall survival (OS) time in the high- and low-risk groups. The OS was significantly shorter in the high-risk group than in the low-risk group (Figure 3A). As the risk score increases, the mortality rate increase (Figure 3B). The risk scores for the high- and low-risk groups are shown in (Figure 3C).

The overall data were then randomly divided into two groups: train and test, which were then subdivided into the high- and low-risk groups according to the median value of the risk score of the training group, and the data of the test group were used for verification. We found that the data of the test (Figure 3D–F) and train groups (Figure 3G–I) were consistent with the trends of the overall data.

Progression-free survival analysis is another way to validate the model. The results show that progression-free survival is better in the low-risk group than in the high-risk group. (Figure 4A). Next, the relationship between the predictive characteristics and prognosis at different clinicopathological stages was verified. The survival of patients in the high-risk groups of stages I–II and III–IV was significantly lower than that of patients in the low-risk groups (Figure 4B,C). The samples were further analyzed by the clinical correlation heat map, and the three sets of data proved that the three protective lncRNAs, AC104088.1, PPP4R3B-DT, and AC006042.3, decreased with an increase in the risk score. In contrast, three risk lncRNAs, LUCAT1, HHLA3-AS1, and LINC02029, increased with an increase in the risk score (Figure 4D–F). The above verification proves the validity of the model.

### 3.3. Independent Analysis of Prognostic Factors

Cox regression analysis is an important method to identify independent prognostic factors that predict characteristics that are effective after radiotherapy in NSCLC patients. Univariate Cox regression analysis proves that OS in patients with non-small cell lung cancer after radiotherapy is closely related to the risk score(*p* < 0.01) (Figure 5A). Multivariate Cox regression analysis proved that the risk score could be used as an independent predictor of OS in patients with NSCLC after radiotherapy (*p* < 0.01) (Figure 5B). The AUCs for 1-, 3-, and 5-year survival were 0.678, 0.792, and 0.806, indicating a good predictive effect (Figure 5C). The concordance index demonstrated that the risk score predicted performance. It was superior to the age, gender, and stage variables of patients in NSCLC after radiotherapy (Figure 5D).

### 3.4. Nomogram and PCA Predictive Construction

The nomogram is a way to predict survival status in patients with NSCLC, so a nomogram including age, gender, stage, and risk scores was constructed. This nomogram predicts 1-, 3-, and 5-year prognoses. (Figure 6A) The calibration curve showed that the proposed model was similar to the ideal model (Figure 6B).

The PCA graph below shows that the Risk lncRNAs (Figure 7A) could better cluster two groups compared to that of the all gene (Figure 7B), cuproptosis gene (Figure 7C), and cuproptosis LncRNA groups (Figure 7D).

### 3.5. Immune Correlation Analysis

The combination of radiotherapy and immunotherapy can effectively enhance therapeutic effects [17,21]. Immune checkpoint analysis proved that the gene expression levels of *CD276* and *TNFSF14* were higher in the high-risk group (Figure 8A). Correlation analysis of the immune escape showed significant differences between groups (*p* < 0.05) (Figure 8B). We performed an immune correlation analysis using multiple software packages. The TIMER software analysis proved that the expression of myeloid dendritic cells was higher in the high-risk group. The CIBERSORT software analysis proved that γ-delta T-cells and M0 macrophages were significantly different between the two groups. The CIBERSORT-ABS software analysis showed significant differences in the γ-delta T-cells and M0 and M2 macrophages. The QUANTISEQ results showed that monocytes and neutrophils were highly expressed in the low-risk group. The MCPCOUNTER analysis showed significant differences in monocytes, macrophages, endothelial cells, and cancer-associated fibroblasts. The XCELL results showed significant differences in CD8+ T-cells and monocytes. The EPIC analysis showed significant differences in cancer-associated fibroblasts, CD8+ T-cells, endothelial cells, and uncharacterized cells (Figure 8C). Among them, M0 macrophages, monocytes, and γ-delta T-cells showed significant differences in multiple prediction software, which deserves further attention and analysis.

### 3.6. Mutation Correlation Analysis

The mutational burden caused by radiation therapy, which can cause mutations in cancer and healthy cells by damaging DNA, is often overlooked during treatment [22]. High and low mutational burdens are involved in the antitumor effects of immune checkpoint inhibitors. Tumors roughly have two immune phenotypes: hot tumors (immune-inflamed) and cold tumors (immune-excluded and immune-desert phenotypes). Hot tumors are characterized by a high tumor mutation burden, which enhances T-cell responses resulting in tumor killing. Conversely, cold tumors show low mutational burdens [23,24,25]. Mutation detection was performed on samples from both groups. Almost all samples in the high- and low-risk groups were mutated, with TP53 being the most common mutation. The gene mutation rate of the low-risk group was 75% higher than that of the high-risk group, according to a comparison of the 50 genes with the highest frequency of mutations. Between the two groups, there were 10% differences in the rates of mutation for *KEAP1, MUC17, RIMS2, FAM135B, ZNF536, PCDH15, ZNF804A, CDH10, CSMD3*, and *MUC16* (Figure 9A,B). The survival analysis results showed that patients with more mutations had greater survival rates. Patients with high mutations and low risk among them had the highest chance of surviving, whereas those with low mutations and high risk had the lowest chance (Figure 9C,D). Some negative effects of cancer treatment, such as local recurrence, could be explained by the sort of mutation radiation therapy causes. Therefore, by converting cold tumors into hot tumors, these genes may be exploited as prospective targets for further investigation in future cuproptosis investigations.

## 4. Discussion

The two most common types of lung cancer are small-cell lung cancer and non-small-cell lung cancer. Squamous cell carcinomas, big cell carcinomas, and adenocarcinomas are the three histological subtypes of NSCLCs. Adenocarcinoma makes up roughly 50% of them, whereas squamous cell carcinoma makes up about 40% of the total [26,27]. More than 85% of cases are currently classified as NSCLC, with a 15.9% estimated 5-year survival rate [28]. The 5-year survival rate has only modestly increased despite continual treatment plan improvement. Radiotherapy combined with the expression of cuproptosis-related lncRNAs might increase patients’ 5-year survival rates. Cuproptosis has a complex role in cancer formation. Cuproptosis has been demonstrated to have an important part in the emergence and growth of tumors in a growing number of studies; nevertheless, current research focuses mostly on the impact of apoptosis in cancer therapy. Some studies have shown that radiotherapy can induce a decrease in the copper metabolism MURR1 domain protein 1 (*COMMD*), which in turn causes an increase in intracellular copper levels, leading to radioresistance [29]. However, the expression and predictive ability of cuproptosis-related lncRNAs in NSCLC patients after radiotherapy need to be further explored. Our clinical application value for radiotherapy and cuproptosis-related lncRNAs is mainly due to the high mortality rate of lung cancer. This is closely related to factors such as late diagnosis, lack of individualized treatment, and resistance to radiotherapy. Early detection and effective combination therapy can improve the efficacy of lung cancer treatment. LncRNAs play important roles in non-small cells [30,31], mediating many cellular processes, including epigenetic regulation and exerting a regulatory effect on mRNA, which can affect mRNA expression and splicing [32]. In NSCLC, most targeted adjunctive sensitization is prone to drug resistance. There is an urgent need to identify sensitive and specific biomarkers associated with radiotherapy, and lncRNAs can be applied as potential biomarkers and therapeutic targets for radiotherapy prognosis. Therefore, it is important to identify lncRNA predictive signatures associated with cuproptosis in NSCLC patients after radiotherapy.

In terms of biomarkers for cancer diagnosis, studies have shown that the upregulation of MALAT-1 in peripheral blood can reflect the presence of NSCLC with a specificity of up to 96% [33]. However, measuring lncRNA levels in peripheral blood alone can lead to inaccuracies; generally, lncRNA levels in the blood are lower due to RNAase degradation, while those in tissues are higher. Therefore, to improve the sensitivity and diagnostic performance of lncRNA detection, several joint lncRNA diagnostic methods may be applied [34,35]. In addition, a series of new technologies for detecting lncRNAs, such as “SPCE Au NCs/MWCNT-NH2”, has paved the way for the well screen of lncRNAs in humans [36]. In terms of predicting cancer prognosis, studies have shown that some lncRNAs can distinguish between metastatic and non-metastatic cancers and benign and malignant tumors. Therefore, the analysis of lncRNAs may help assess disease progression. In addition, lincRNA-p21 has been described as a tumor suppressor, while HOTAIR, MALAT-1, H19, PVT1, ANRIL, and GIHCG, have been characterized as oncogenic lncRNAs. Therefore, our studies have shown the identification of lncRNAs associated with cuproptosis in patients with NSCLC undergoing radiotherapy is of great significance for early diagnosis, combination therapy, prognostic analysis, and reduction of radiation resistance.

LncRNAs play an important regulatory role in NSCLC radiotherapy. lncRNA *CRNDE/PRC2* targeting p21 sensitized radioresistance in NSCLC [37]; SPRY3-2/3/4 and H19–miR-130a-3p–WNK3 axis modulates radiosensitivity and affects apoptosis and cell viability in NSCLC [38,39]. However, lncRNAs associated with radiotherapy-related cuproptosis have never been studied in NSCLC. Our studies have shown that LUCAT1 is significantly upregulated in NSCLC, where it suppresses the expression of p21 and p57 by associating with PRC2 to mediate epigenetic regulation. The development of cisplatin resistance in NSCLC is closely related to autophagy, affects tumor proliferation, and is closely linked to patient prognosis [40,41,42]. HHLA3 is an important member of the clear cell renal cell carcinoma cuproptosis-related prognostic model [43] and is tightly related to the metabolic immune infiltration of NSCLC [44]. The remaining lncRNAs described here were reported for the first time. In particular, these newly discovered NSCLC-related lncRNAs may serve as new targets for cancer therapy.

Currently, the antitumor mechanism by cuproptosis is unclear. This study showed that cuproptosis occurs predominantly due to the accumulation of intracellular copper, which triggers the accumulation of mitochondrial fatty acylated proteins and the instability of Fe-S cluster portions, which induces a type of cell death that is different from oxidative stress-related cell death [45]. Elevated copper concentrations have been found in patients with various cancers. Therefore, copper chelators and nanodelivery modalities are expected to be developed as adjuvant therapies for tumors in the future. We may also imagine that the use of nanodelivery could improve the efficacy of cuproptosis-mediated treatment. Nanodelivery could be used for designing Cu-based and non-Cu-based nanodrug delivery systems (NDDSs). A prior study showed that ten regulated genes were specifically related to the cuproptosis metabolic pathway, including seven positively regulated genes and three negatively regulated genes [10]. The resulting co-expression network indicated that four of our six key predicted genes were co-expressed with the negative regulator genes, *CDKN2A* and *GLS*. Therefore, the key regulatory genes in NSCLC patients after radiotherapy need to be explored further.

The effective blockade of immune checkpoints can activate innate immune pathways by increasing DNA damage, thereby enhancing radiosensitivity [46,47,48]. Significant differences in *CD276* and *TNFSF14* expression were observed between the high- and low-risk groups. Studies have shown that *CD276* CAR T-cells can reduce radiation resistance in prostate cancer by targeting prostate cancer stem cells [49]. Similarly, *TNFSF14* profoundly affects immune responses within tumors, not only sensitizing cells to IFNγ-mediated apoptosis [50] but also stimulating effector cell function and antitumor CD8+ T-cell entry into tumors [51,52,53,54]. These two immune checkpoints may provide new research ideas for cuproptosis combined with radiotherapy.

The three predominant predictive biomarkers of PD-1 blockade therapy are tumor mutational burden, CD8+ T-cell infiltration intensity, and programmed cell death ligand 1 (PD-L1) expression [55]. Among these, DNA repair defects are associated with increased mutational loads [56]. Several studies have shown that mutations in TP53 in lung adenocarcinoma increase sensitivity to the PD-1 blockade [55]. Moreover, the *KEAP1/NRF2* mutation status of patients with NSCLC can affect local recurrence after radiotherapy through tumor invasion and metastasis [57]. Therefore, when radiotherapy combined with immunotherapy is performed for high-risk groups, analysis of the mutated genes can provide a theoretical basis for the precise and individualized treatment of NSCLC patients.

Our research still has certain limitations. First, data from the other databases were needed for external validation to check the applicability of the prediction model because only the TCGA database was utilized for internal validation via random grouping. Second, more experimental research is required into the mechanism of cuproptosis-related lncRNAs in NSCLC.

In conclusion, cuproptosis-related lncRNA characteristics can independently predict a patient’s prognosis, provide evidence for a potential mechanism underlying these lncRNAs in NSCLC, and predict the effectiveness of a patient will respond to clinical treatment; however, more research is required to validate these findings.

## Figures and Tables

**Figure 1 genes-13-02080-f001:**
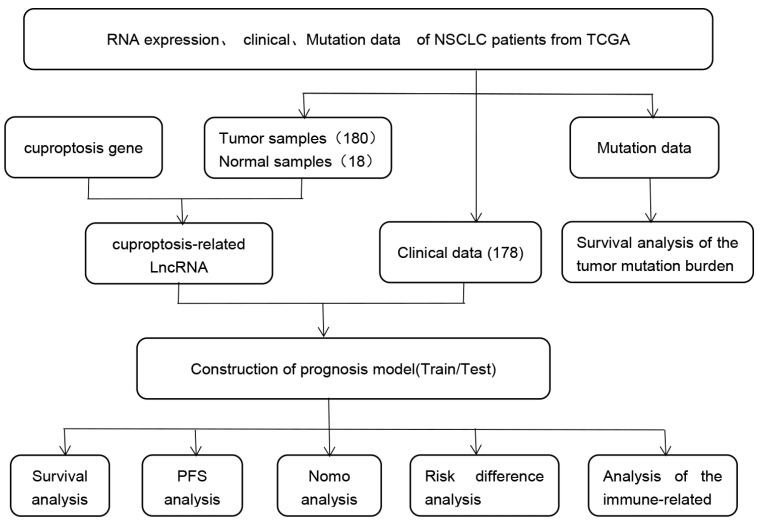
NSCLC, Non-small-cell carcinoma; TCGA, The Cancer Genome Atlas; PFS, progression-free survival; LncRNAs, long noncoding RNAs.

**Figure 2 genes-13-02080-f002:**
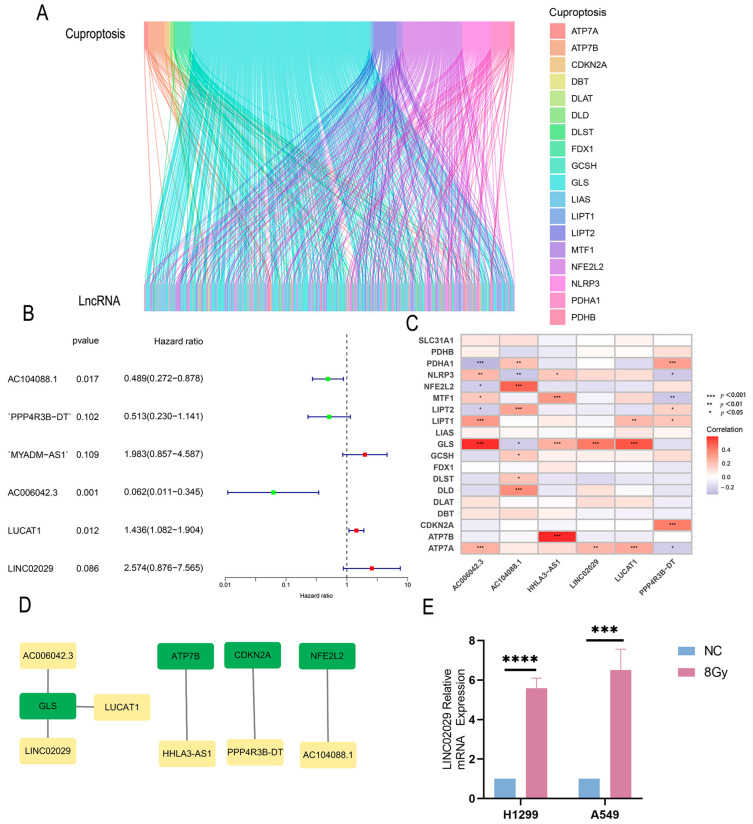
(**A**) Sankey diagram for the co-expression network in NSCLC. The above axis of the Sankey diagram represents cuproptosis-related genes, while the below axis represents the related lncRNA. (**B**) Hazard ratios (95% CI) of the different lncRNA by cox regression. Red represents a high-risk HR > 1; green represents a low-risk HR < 1. (**C**) Heatmap plots of the differentially expressed lncRNAs. The horizontal axis represents the risk of LncRNA. The vertical axis represents cuproptosis-related genes. The color represents the correlation between the two. Blue indicates a negative correlation, while red indicates a positive correlation. (**D**) Cytoscape software builds the Gene and LncRNA co-expression network, yellow for risk prediction of LncRNA, and green for co-expression of cuproptosis genes. (**E**) Real-time Quantitative PCR analysis of the expression pattern of LINC02029 prognostic lncRNAs in H1299 and A549 cells. * *p* < 0.05, ** *p* < 0.01, *** *p* < 0.001, **** *p* < 0.0001.

**Figure 3 genes-13-02080-f003:**
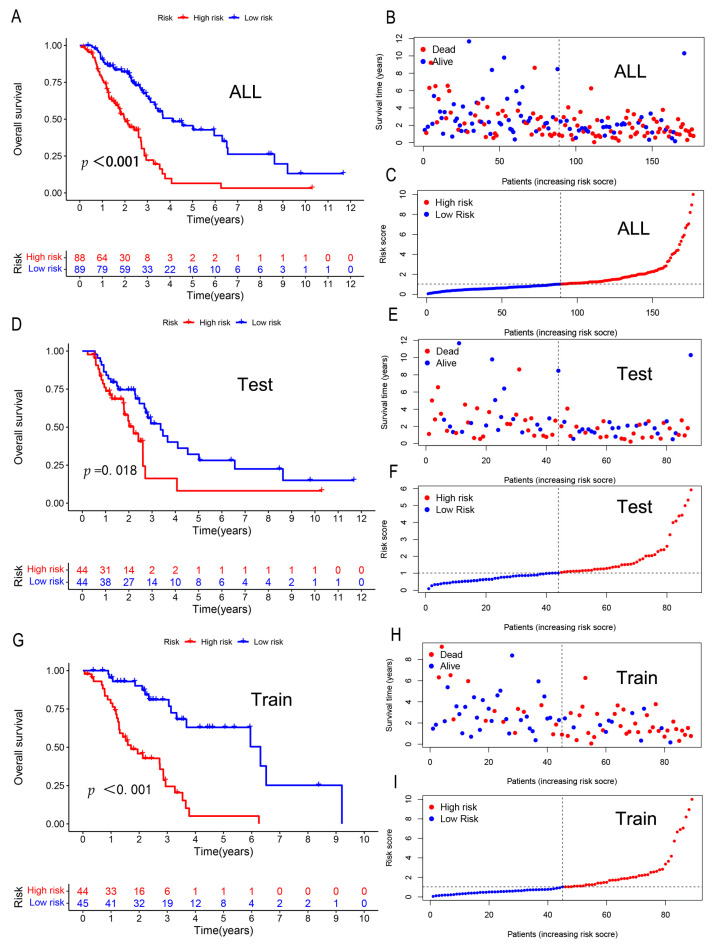
Kaplan–Meier curve evaluation (**A**,**D**,**G**) The overall survival rates of NSCLC patients in various categories. Abscissa: Time; ordinate: survival rate; Each point represents the patient’s survival rate at that moment. (**B**,**E**,**H**) The number of survivors and deaths at various risk levels. The colors blue and red symbolize survival and death, respectively. Survival status curves of patients with varying risk ratings (**C**,**F**,**I**). (**A**–**C**) represent all groups; (**D**–**F**) represent the test group; and (**G**–**I**) represent the trian group.

**Figure 4 genes-13-02080-f004:**
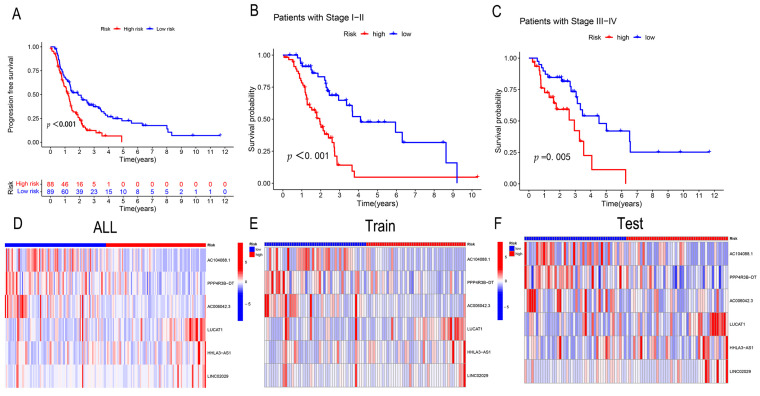
(**A**) The Kaplan–Meier curve for progression-free survival in the high and low-risk groups, with the abscissa representing time and the ordinate indicating progression-free survival. (**B**,**C**) Validation of the clinical grouping models in stages I–II and III–IV. (**D**–**F**) LncRNA risk heat map of 177 samples, with the abscissa representing the high and low-risk groups and the ordinate representing the risk prediction LncRNA.

**Figure 5 genes-13-02080-f005:**
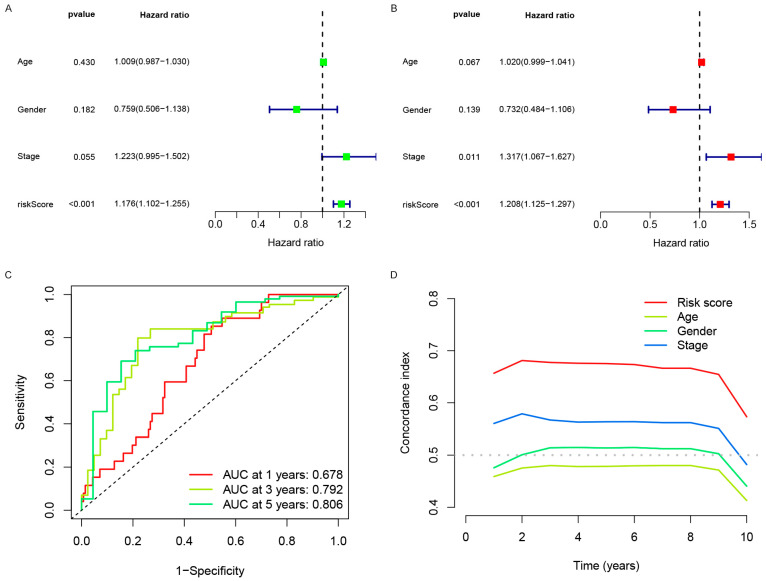
(**A**) forest plot of a univariate Cox regression analysis, green represents Hazard ratio (95% CI). (**B**) Forest plot of the multivariate Cox regression analysis using independent prognostic variables and the Red represents hazard ratio (95% CI). (**C**) The predictive signature’s AUCs for survival. The ordinate is the genuine positive rate annotation method, while the abscissa is the false positive rate annotation method with 1-specificity. Sensitivity is represented by the red, yellow, and green ROC curves, with red denoting one year, yellow three years, and green five years. (**D**) The concordance index of the risk score was evaluated utilizing the C-index curve. The red curve represents the risk score, the yellow curve represents age, the green curve represents gender, and the blue curve represents the stage. The abscissa represents time, the ordinate represents the C-index value, estimated by bootstrapping (B = 1000 repetitions), and the larger the value, the higher the accuracy of the model’s prediction.

**Figure 6 genes-13-02080-f006:**
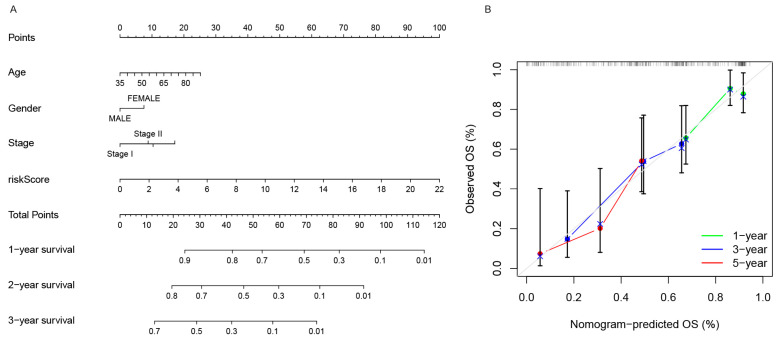
(**A**) Clinicopathological variable nomograms and risk scores to predict 1-, 3-, and 5-year OS in NSCLC patients. (**B**) The nomogram calibration curves. The *x*-axis indicates the nomogram-predicted OS probability, while the *y*-axis represents the NSCLC patients’ observed OS probability. The 45° gray line would equate to a perfect forecast. The green, blue, and red lines reflect the observed nomogram performance at 1, 3, and 5 years, respectively, as assessed by bootstrapping (B = 1000 repetitions).

**Figure 7 genes-13-02080-f007:**
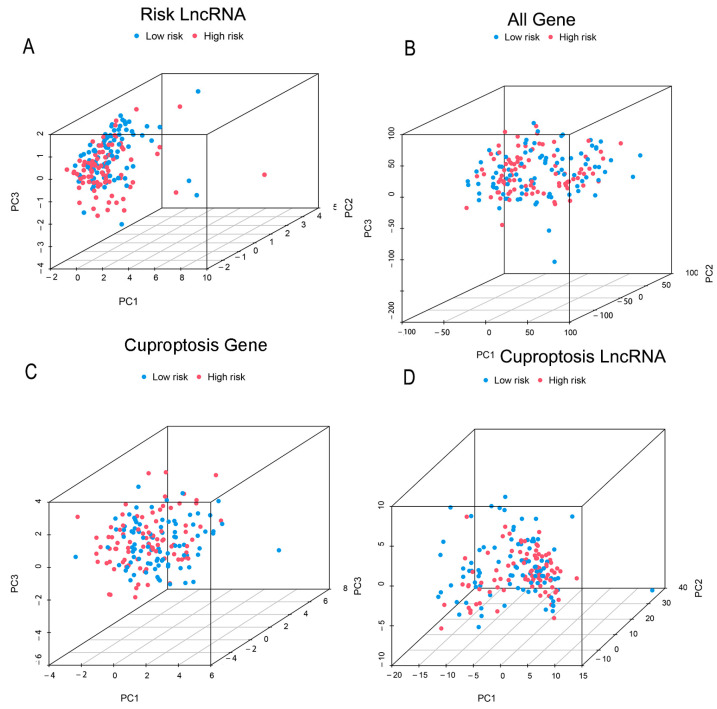
The results of the principal component analysis. PCA analysis of the patient distribution according to (**A**) risk lncRNA, (**B**) all genes, (**C**) cuproptosis-associated genes, and (**D**) cuproptosis-associated lncRNAs.

**Figure 8 genes-13-02080-f008:**
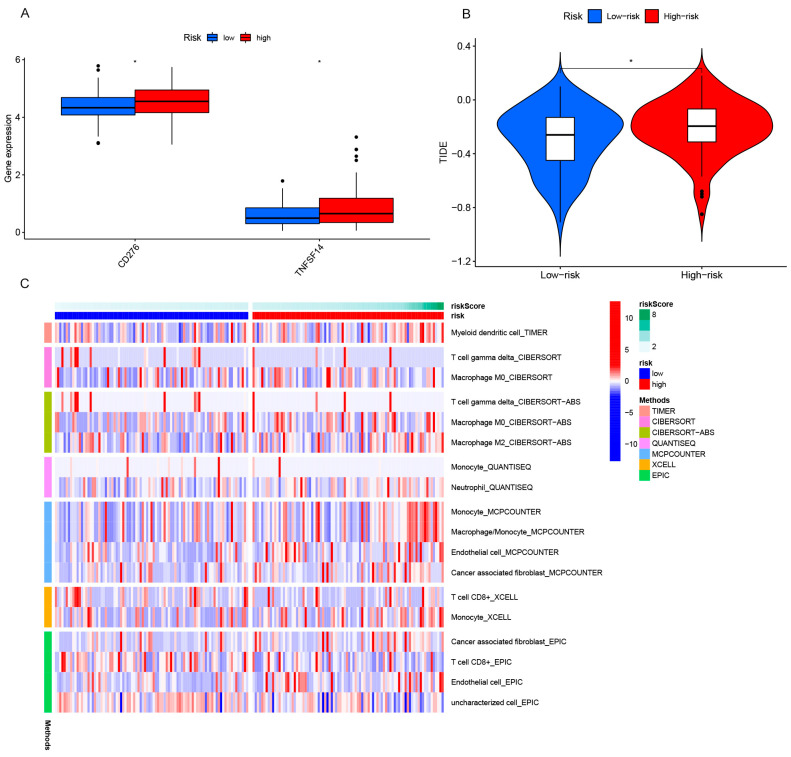
(**A**) Immune checkpoint analysis in high- and low-risk groups, in which the ordinate represents the level of gene expression and the abscissa of the immune checkpoint-related gene. TNFSF14, recombinant Tumor Necrosis Factor Ligand Superfamily, Member 14; CD276, the recombinant cluster of differentiation 276. (**B**) In the TIDE (Tumor Immune Dysfunction and Exclusion) abscissa, the high and low-risk categories are represented by blue and red, respectively. TIDE score * *p* < 0.05 serves as the ordinate. (**C**) Relationship analysis of the immune cells in high- and low-risk groups. The ordinate represents immune cells, the abscissa represents high- and low-risk groups, and the various colors represent the predictions from various software programs.

**Figure 9 genes-13-02080-f009:**
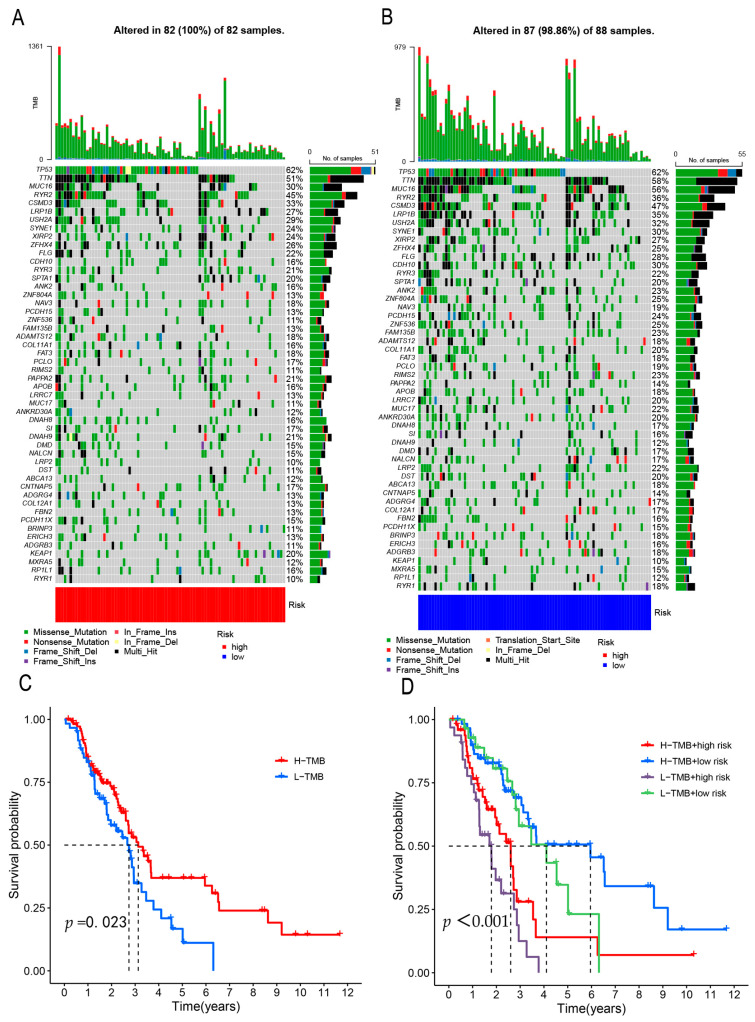
(**A**,**B**) A waterfall plot depicts the top 50 genes in the high and low-risk categories, respectively. Each abscissa represents a single sample, the left ordinate represents the gene, the right ordinate shows the frequency of gene mutation, and the different colors of the legend reflect different mutation types. (**C**,**D**) Tumor mutation burden survival curve, the abscissa represents survival time, the ordinate represents survival rate, red represents the high mutation burden group, and blue represents the low mutation burden group. The dit line stands for half the probability of survival.

**Table 1 genes-13-02080-t001:** Primer sequences for qRT-PCR.

Primer	Sequence
LINC02029-F	TAGAGATGGAGGACTGGGAGG
LINC02029-R	GTGCACACTTGTCCAAGCAG
GAPDH-F	GTCTCCTCTGACTTCAACAGCG
GAPDH-R	ACCACCCTGTTGCTGTAGCCAA

**Table 2 genes-13-02080-t002:** Main clinical and demographic characteristics of NSCLC patients within various datasets.

Variables	Entire TCGA Dataset *n* = 177		Train *n*1 = 88			Test *n*2 = 89
Age (%)						
≤65	90	50.85%	41	46.59%	49	55.06%
>65	84	47.46%	47	53.41%	37	41.57%
Unknown	3	1.69%			3	3.37%
Gender (%)						
Female	85	48.02%	47	53.41%	38	42.70%
male	92	51.98%	41	46.59%	51	57.30%
Stage (%)						
I + II	104	58.76%	49	55.68%	55	61.80%
III + IV	70	39.55%	38	43.18%	32	35.96%
Unknow	3	1.69%	1	1.14%	2	2.25%
T (%)						
T1 + T2	135	76.27%	66	75.00%	69	77.53%
T3 + T4	40	22.60%	21	23.86%	19	21.35%
TX + Unknow	2	1.13%	1	1.14%	1	1.12%
M (%)						
M0	124	70.06%	61	69.32%	63	70.79%
M1	10	5.65%	4	4.55%	6	6.74%
Mx + Unknow	43	24.29%	23	26.14%	20	22.47%
N (%)						
N0	82	46.33%	38	43.18%	44	49.44%
N1 + N2	88	49.72%	45	51.14%	43	48.31%
N3	3	1.69%	3	3.41%		
NX + Unknow	4	2.26%	2	2.27%	2	2.25%

T, tumor; M, metastasis; *n*, lymph node.

## Data Availability

The data presented in this study are openly available in https://portal.gdc.cancer.gov/, accessed on 31 October 2022.

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
