# Peer review of "Radiosensitization-Related Cuproptosis LncRNA Signature in Non-Small Cell Lung Cancer"

_genes, 2022, doi:10.3390/genes13112080_

Round 1
Reviewer 1 Report
In the manuscript Radiosensitization-related Cuproptosis LncRNA signature in non-small cell lung cancer, Qiushi Xu et al. used data from The Cancer Genome Atlas (TCGA) database to find and validate a cuproptosis-related lncRNA signature to predict NSCLC patients prognosis after radiotherapy. In this in silico study, the authors show that copper-related lncRNAs are potential biomarkers for radiosensitization and set the bases for future clinical studies in NSCLC patients.
Major:
As shown in Figure 1, the authors performed multiple analyses to construct the prognosis model (Survival analysis, PFS analysis, Nomo analysis, risk difference analysis, and analysis of the immune-related). The methods section does not adequately describe the methodology used to perform these analyses. Furthermore, the construction of the nomogram is the only analysis addressed in the methods section, and, with the information provided, it is unclear which tools were used to elaborate and validate it. Did the authors validate the predictive performance for discrimination on the original cohort using bootstrapping? In line 62, the authors mention they used Cytoscape software to analyze co-expression, which is also not described in the methods section.
In Figure 2A, the authors show a co-expression relationship in 198 TCGA samples with cuproptosis-related genes. However, it is unclear how these genes (ATP7A, ATP7B, CDKN2A, DBT, DLAT, DLD, DLST, FDX1, GCSH, GLS, LIAS, LIPT1, LIPTR, MTF1, PDHA1, NLRP3, and PDHB) were selected. Could the authors elaborate on why they found and validated a lncRNA signature associated with these cuproptosis genes rather than just using these same cuproptosis genes as predictors? Is there a reason why a lncRNA signature would be a superior predictor of prognosis? Also, the description of Figure 2A in the text (lines 131-140) is hard to follow; thus, a table would be more suitable to describe all the co-expression relationships.
Figures 2 E and F quantify the expression of LINC02029 in A549 and H1299 cell lines after radiotherapy. However, the conditions in which these experiments were performed (radiation doses, time of exposure, type of radiation applied) are not reported within the manuscript. Also, the color-coded legends in these figures are redundant, and the lncRNA name is misspelled (LINCO2029 -> LINC02029).
The focal point in the manuscript is the signature with six lncRNAs that can predict prognosis in the data analyzed. However, the known functions of these lncRNAs are not discussed despite their importance. For example, is there evidence of their role in cuproptosis or NSCLC?
Minor:
Define LncRNA and NSCLC when they first appear, and be consistent with the abbreviations.
Line 13: remove Novel cuproptosis Predictive Signature in Non-Small Cell Lung Cancer After Radiotherapy.
Line 72: LncRNA MIVH -> LncRNA associated with microvascular invasion in hepatocellular carcinoma (MIVH)
Lines 83, 86, 87: specify the dates in which the databases https://portal.gdc.cancer.gov/, http://timer.cistrome.org, and http://tide.dfci.harvard.edu/ were accessed to download information.
Lines 91-94: the results from the initial screening, 535 cuproptosis-related LncRNAs, and the final six LncRNAs used to construct a cuproptosis-related predictive signature associated with NSCLC prognosis should be listed in a supplementary file.
Line 109: For qRT-PCR, the thermocycler used is not mentioned
Line 170: qpCR -> qPCR
Line 170: remove As shown in figure,
Line 265: fibroblast -> fibroblasts
Lines 279-281: add references
Line 317: the word blank is unclear; please rephrase
Author Response
We thank this reviewer for the insightful comments. We agree with this reviewer that our studies of Radiosensitization-related Cuproptosis LncRNA can be further strengthened,and hope that our revision now has largely addressed the critiques from this reviewer.
Major:
As shown in Figure 1, the authors performed multiple analyses to construct the prognosis model (Survival analysis, PFS analysis, Nomo analysis, risk difference analysis, and analysis of the immune-related). The methods section does not adequately describe the methodology used to perform these analyses. Furthermore, the construction of the nomogram is the only analysis addressed in the methods section, and, with the information provided, it is unclear which tools were used to elaborate and validate it. Did the authors validate the predictive performance for discrimination on the original cohort using bootstrapping? In line 62, the authors mention they used Cytoscape software to analyze co-expression, which is also not described in the methods section.
We thanked the insightful question of the reviewer. About this question, We have made detailed changes in the methodology section, which are as follows
Materials and methods
Data sources
The RNA-Seq dataset and corresponding clinical follow-up information of non-small cell lung cancer after radiotherapy were downloaded from the TCGA database (https://portal.gdc.cancer.gov/) on June 10, 2022, obtained 180 primary tumors and 18 solid tissue normal data were collected. Clinical data of 178 patients who underwent radiation therapy.
.Cuproptosis-related genes were collected through the literature search.
Inclusion and exclusion criteria
Inclusion criteria: ①Bronchi and lung were screened in the database; ②Squamous cell carcinoma and adenocarcinoma were screened by disease type; ③Primary tumor and solid normal tissue were screened by sample type; The experimental protocol was transcriptome analysis.
Exclusion criteria: ① The data information is incomplete.
Construction of predictive signatures of cuproptosis-related lncRNAs
The “limma”package was used to calculate the correlation between cuproptosis-related genes and lncRNAs. Using correlation coefficient corFilter=0.4 and pvalueFilter<0.001 as screening criteria, a total of 535 cuproptosis-related lncRNAs were obtained(Supplementary table S1)T. Further use logFCfilter=1, fdrFilter=0.05 to obtain the differential lncRNA between normal tissue and tumor tissue. Differential lncRNAs were obtained using univariate Cox regression analysis to obtain lncRNAs associated with the prognosis of patients with non-small cell lung cancer, and then multivariate Cox analysis was performed on the selected genes, and then the stepwise regression method (R package stepAIC) was used to further reduce the number of genetic risk factor. Finally, a prognostic model is constructed using the selected genes. The calculation formula used for this analysis is as follows:
Risk Score = x1∗coef1 x2∗coef2 … xn∗coefn
Coef represents the coefficient value, and x represents the expression value of selected cuproptosis-related lncRNAs. This formula was used to calculate the risk score for each patient with non-small cell lung cancer after radiotherapy. The patients were divided into high and low risk groups according to the median of the risk score.
Survival analysis and progression-free survival(PFS) analysis
In order to verify the survival difference between high and low risk groups, the TGCA data were divided into training set and validation set. In order to avoid the deviation of random assignment, all samples were randomly divided into 100 groups without replacement. The training dataset and validation dataset is 1 versus 1. Use the “survival” package and the〝survminer "package to visualize different groupings respectively. (https://xenabrowser.net) to download the pan-cancer clinical data literature (Supplementary Table S2). The difference in PFS was validated using the“survival” package and the “survminer ” package.
co-expression network
Co-expression analysis of cuproptosis gene expression data and LncRNA expression evidence was performed using the “limma” package. Using correlation coefficient corFilter=0.4 and pvalueFilter=0.001 as filter criteria, cytoscape software analyzed cuproptosis-related genes and LncRNAs involved in model construction, and obtained a prognosis-related co-expression network map.
Nomo and Calibration plots
Combining risk scores with clinical information on age, sex, stage, and N , nomogram survival models for 1, 3, and 5year OS were built using the R package” rms”, based on univariate and multivariate outcomes. The calibration curve verifies the performance of the nomogram; the calibration curve estimate was adjusted for optimism by applying a bootstrap(B=1000 repetitions) procedure.
Tumor Immunoassay
To explore the relationship between risk models and immune infiltration status, immune-related files were downloaded. Immune infiltrating cell file download address (http: timer.comp-genomics.org)on June 18, 2022. Immune escape file download address (http://tide.dfci.harvard.edu/)on June 23, 2022, using “limma”package and “pheatmap” package to analyze the differences between high and low risk groups.
Tumor mutational burden analysis
The mutation data of non-small cell lung cancer after body radiotherapy was downloaded from the TCGA website, and the Perl programming language (version strawberry-Perl-5.30.0; https://www.perl.org) was used to obtain the mutation data. The 50 genes with the highest mutation frequency were visualized using the “maftools”package. Tumor mutational burden was subjected to survival analysis using the”maftools”package
In Figure 2A, the authors show a co-expression relationship in 198 TCGA samples with cuproptosis-related genes. However, it is unclear how these genes (ATP7A, ATP7B, CDKN2A, DBT, DLAT, DLD, DLST, FDX1, GCSH, GLS, LIAS, LIPT1, LIPTR, MTF1, PDHA1, NLRP3, and PDHB) were selected. Could the authors elaborate on why they found and validated a lncRNA signature associated with these cuproptosis genes rather than just using these same cuproptosis genes as predictors? Is there a reason why a lncRNA signature would be a superior predictor of prognosis?
We thanked the insightful question of the reviewer. About this question,Cuproptosis-related genes were collected through the literature search.and The discussion on this issue is supplemented below.
Our clinical application value for radiotherapy and cuproptosis -related LncRNA is mainly due to the high mortality rate of lung cancer. This is closely related to factors such as failure to diagnose early, lack of individualized treatment, and resistance to radiotherapy. Early detection and effective combination therapy contribute to the efficacy of lung cancer. Radiation therapy is indicated at all stages of non-small cell lung cancer, and approximately 77 percent of patients with lung cancer should receive radiation therapy [22,23]. To enhance the effectiveness of radiation therapy, a high radiation dose is often required, which can lead to unacceptable side effects. Most targeted adjunctive sensitization is prone to drug resistance. Therefore, there is an urgent need to identify sensitive and specific biomarkers associated with radiotherapy, and lncRNA can be used as potential biomarkers and therapeutic targets for the prognosis of radiotherapy. In terms of biomarkers for cancer diagnosis, studies have shown that the upregulation of MALAT-1 in peripheral blood can reflect the presence of non-small cell lung cancer with a specificity of up to 96% [27]. However, measuring LncRNA levels in peripheral blood alone can lead to lower levels in blood and higher levels in tissues due to RNAse degradation. Therefore, in order to improve the sensitivity and diagnostic performance of LncRNA detection, several LncRNA joint diagnostic methods can be selected [28][29]. In addition, a series of new technologies for detecting LncRNAs, such as "SPCE Au NCs/MWCNT-NH2", have paved the way for better detection of LncRNA in humans [30]. In terms of predicting cancer prognosis, studies have shown that some lncRNAs can distinguish between metastatic and non-metastatic cancers, as well as benign and malignant tumors. Therefore, the level of LncRNA may help assess the progression of the disease. In addition, lincRNA-p21 has been described as a tumor suppressor, while MALAT-1, HOTAIR, H19, PVT1, GIHCG and ANRIL have been characterized as oncogenic lncRNAs. Therefore, the identification of lncRNAs associated with cuproptosis in patients with non-small cell lung cancer in radiotherapy is of great significance for early diagnosis, combination therapy, prognostic analysis, and radiation resistance reduction.
Also, the description of Figure 2A in the text (lines 131-140) is hard to follow; thus, a table would be more suitable to describe all the co-expression relationships.
We thanked the insightful question of the reviewer. About this question, We have made changes , which are as follows
The co-expression relationship were visualized using a Sankey diagram ,The data are presented in supplementary table S3.
Figures 2 E and F quantify the expression of LINC02029 in A549 and H1299 cell lines after radiotherapy. However, the conditions in which these experiments were performed (radiation doses, time of exposure, type of radiation applied) are not reported within the manuscript. Also, the color-coded legends in these figures are redundant, and the lncRNA name is misspelled (LINCO2029 -> LINC02029).
We thanked the insightful question of the reviewer. About this question, We have addde to the content , which are as follows
We applied X-ray at a dose rate of 6 Gy/min, continued the cell dose of 0 Gy、 8 Gy, respectively, collected cells for qpCR 48 hours after radiotherapy.
After literature research lncRNA name may be ( LINC02029 ->LINCO2029)?
The focal point in the manuscript is the signature with six lncRNAs that can predict prognosis in the data analyzed. However, the known functions of these lncRNAs are not discussed despite their importance. For example, is there evidence of their role in cuproptosis or NSCLC?
We thanked the insightful question of the reviewer. About this question,Additions were made in the discussion.
In these LcnRNAs, studies have shown that LUCAT1 is significantly upregulated in NSCLC, and LUCAT1 represses the expression of p21 and p57 via associating with PRC2 to regulate epigenetics. Develops cisplatin resistance in NSCLC. It is closely related to NSCLC autophagy, affects the proliferation of tumors, and is closely related to the prognosis of patients. [28] [29] [30] HHLA3 is an important member of the clear cell renal cell carcinoma cuproptosis-related prognostic model [31] and is closely related to the metabolic immune infiltration of NSCLC. [32] The remaining LncRNAs are reported for the first time. In particular, these newly discovered NSCLC-related cuproptosis-related lncRNAs may be new targets for cancer therapy.
Minor:
Define LncRNA and NSCLC when they first appear, and be consistent with the abbreviations.
We thanked the insightful question of the reviewer. About this question, We have made change , which are as follows
However, the predictive value of cuproptosis-related Long non-coding RNA(LncRNA)in non-small cell lung cance(NSCLC)after radiotherapy remains to be further elucidated.
Line 13: remove Novel cuproptosis Predictive Signature in Non-Small Cell Lung Cancer After Radiotherapy.
We thanked the insightful question of the reviewer. About this question, We have removed the sentence.
Line 72: LncRNA MIVH -> LncRNA associated with microvascular invasion in hepatocellular carcinoma (MIVH)
We thanked the insightful question of the reviewer. About this question, We have made change , which are as follows
LncRNA associated with microvascular invasion in hepatocellular carcinoma (MIVH) regulates the proliferation and invasion of squamous cell carcinoma by affecting the expression of MMP-2/-9.
Lines 83, 86, 87: specify the dates in which the databases https://portal.gdc.cancer.gov/, http://timer.cistrome.org, and http://tide.dfci.harvard.edu/ were accessed to download information.
We thanked the insightful question of the reviewer. About this question, We have made change , which are as follows
The RNA-Seq dataset and corresponding clinical follow-up information of non-small cell lung cancer after radiotherapy were downloaded from the TCGA database (https://portal.gdc.cancer.gov/) on June 10, 2022
To explore the relationship between risk models and immune infiltration status, immune-related files were downloaded. Immune infiltrating cell file download address (http: timer.comp-genomics.org)on June 18, 2022. Immune escape file download address (http://tide.dfci.harvard.edu/)on June 23, 2022
Lines 91-94: the results from the initial screening, 535 cuproptosis-related LncRNAs, and the final six LncRNAs used to construct a cuproptosis-related predictive signature associated with NSCLC prognosis should be listed in a supplementary file.
We thanked the insightful question of the reviewer. About this question, We have made a supplementary file.
Line 109: For qRT-PCR, the thermocycler used is not mentioned
We thanked the insightful question of the reviewer. About this question, We have addde to the content , which are as follows
Thermo Fisher Scientific Applied Biosystems QuantStudio5 real-time PCR system detects the expression level of LNCRNA-LINC02029
Line 170: qpCR -> qpCR
We thanked the insightful question of the reviewer. About this question, We have made change ,
Line 170: remove As shown in figure,
We thanked the insightful question of the reviewer. About this question, We have removed the sentence .
Line 265: fibroblast -> fibroblasts
We thanked the insightful question of the reviewer. About this question, We have made changed.
Lines 279-281: add references
We thanked the insightful question of the reviewer. About this question, We have added references .
Figure 8. (A) CD276, Recombinant Cluster Of Differentiation 276[23]; TNFSF14, Recombinant Tumor Necrosis Factor Ligand Superfamily, Member 14[24]. (B) TIDE, Tumor Immune Dysfunction and Exclusion, *p<0.05. (C) Methods for different immune prediction software
Line 317: the word blank is unclear; please rephrase
We thanked the insightful question of the reviewer. About this question, We have changed the sentence to :However, the prediction of cuproptosis-related LncRNAs in non-small cell lung cancer patients after radiotherapy needs to be further explored.

Reviewer 2 Report
Dear authors,
Congratulations on the performed hard work to prepare the manuscript.
Your research provides support for the function of cuproptosis-related LncRNAs in non-small cell lung cancer after radiotherapy by revealing a risk model that may independently predict the prognosis of patients with non-small cell lung cancer after radiotherapy. It has been established that cuproptosis, a type of cell death distinct from oxidative stress-related cell death, is primarily caused by intracellular copper buildup, which causes the aggregation of mitochondrial fatty acylated proteins and the instability of Fe-S cluster proteins.
In the manuscript, data are well presented in corresponding figures, tables with information about statistical significance. The materials and methods are easy to understand and support the base for the results.
The obtained results were discussed in relation to the previously published data and based on this study's novelty, which is clearly emphasized. Moreover, study limitations are clearly stated, and further research is needed in the field.
The last 2 words from conclusions (“Experimental verification.”) should be deleted or rephrased into a sentence.
Kind regards,
Author Response
We thank this reviewer for the insightful comments. We agree with this reviewer that our studies of Radiosensitization-related Cuproptosis LncRNA can be further strengthened,and hope that our revision now has largely addressed the critiques from this reviewer.
The last 2 words from conclusions (“Experimental verification.”) should be deleted or rephrased into a sentence.
We thanked the insightful question of the reviewer. About this question, We have deleted the 2 words.
Reviewer 3 Report
In this study, authors show a novel cuproptosis predictive signature in non-small cell lung cancer (NSCLC) patients after radiotherapy. This is an interesting and well conducted study but there are some issues:
Major issues
1) Authors should improve the Materials and Methods section since more details are needed to easily understand how they carried out the study.
2) Under my point of view, an extensive revision of English should be performed because I have found several grammatical mistakes in all sections of the text. For example several phrases were difficult to understand (Example in Material and Methods: “We downloaded normalized RNA-seq data for radiotherapy-treated NSCLC from the TCGA website (https://portal.gdc.cancer.gov/), along with corresponding clinical and mutational data; 180 primary tumors and were obtained Example solid tissue normal data. Clinical data of 178 patients who underwent radiation therapy. (http://timer.cistrome.org) website to download information on immune infiltration. (http://tide.dfci.harvard.edu/) website to download data on immune escape”).
3) Authors should include more details in figure legends since descriptions of figures are very simple, and in some cases difficult to understand.
4) The “Conclusions” sections should be renamed as “Discussion”. In this respect, authors have referenced others studies in this section to discuss their findings. I think that they should included 1 or 2 paragraphs to discuss their own findings because of the novelty of their study.
Minor issues
1) Statistical analyses performed should be showed in all figure legends.
Author Response
We thank this reviewer for the insightful comments. We agree with this reviewer that our studies of Radiosensitization-related Cuproptosis LncRNA can be further strengthened,and hope that our revision now has largely addressed the critiques from this reviewer.
Major issues
1) Authors should improve the Materials and Methods section since more details are needed to easily understand how they carried out the study.
We thanked the insightful question of the reviewer. About this question, We have revised this part in the methodology section, which are as follows
Materials and Methods
Data sources
The RNA-Seq dataset and corresponding clinical follow-up information of NSCLC after radiotherapy were downloaded from the TCGA database (https://portal.gdc.cancer.gov/) on June 10, 2022. We collected the data of 180 primary tumors and 18 normal solid tissue samples,the clinical data of 178 patients who underwent radiation therapy . Furthermore,the data on the cuproptosis-related genes were collected through a literature search.[10][18][19][20]
Inclusion and exclusion criteria
The inclusion criteria were as follows: 1. bronchi and lungs were screened in the database; 2. squamous cell carcinoma and adenocarcinoma were screened by disease type; 3.primary tumor and solid normal tissue were screened by sample type; and 4. transcriptome analysis was performed.
The exclusion criterion was incomplete data.
Construction of predictive signatures with cuproptosis-related lncRNAs
The “limma” package was used to calculate the correlation between cuproptosis-related genes and lncRNAs. Using the correlation coefficient corFilter=0.4 and p-value Filter < 0.001 as the screening criteria, a total of 535 cuproptosis-related lncRNAs were identified(Supplementary table S1). The cutoff limits LogFCfilter =1 and fdrFilter=0.05 were used to define differential lncRNAs between normal and tumor tissue. Differential lncRNAs were identified using univariate Cox regression analysis to identify lncRNAs associated with the prognosis of patients with NSCLC, and then multivariate Cox analysis was performed on the selected genes. A stepwise regression method (R package stepAIC) was used to further reduce the number of genetic risk factors. Finally, a prognostic model was constructed using these selected genes. The formula used for this analysis was as follows:
Risk Score = x1∗coef1 x2∗coef2 … xn∗coefn
Here, Coef represents the coefficient value, and x represents the expression value of the selected cuproptosis-related lncRNAs. This formula was used to calculate the risk score for each patient with NSCLC after radiotherapy. Patients were divided into high- and low-risk groups according to the median risk score.
Survival analysis and progression-free survival (PFS) analysis
To verify the survival difference between the high- and low-risk groups, TGCA data were divided into the training and validation sets. To avoid deviation from random assignment, all samples were randomly divided into 100 groups without replacement. The training and validation datasets were matched 1 to 1. We used the “survival” package and the “survminer” package to visualize different groupings, and pan-cancer clinical data were downloaded from https://xenabrowser.net (Supplementary Table S2). The difference in PFS was validated using the Kaplan-Meier “survival” package and the “survminer ” package.
Co-expression network
Co-expression analysis of the cuproptosis gene expression data and lncRNA expression was performed using the “limma” package. Using the correlation coefficient corFilter=0.4 and p-value Filter=0.001 as the filter criteria, the Cytoscape software was used to analyze the cuproptosis-related genes and lncRNAs involved in model construction. A prognosis-related co-expression network map was thereby constructed.
Nomo and Calibration plots
Combining risk scores with clinical information on age, sex, and stage, nomogram survival models for 1-, 3-, and 5-year OS were built using the R package ”rms,” based on univariate and multivariate outcomes. The calibration curve verified the performance of the nomogram, and the calibration curve estimate was adjusted for optimism by applying a bootstrap procedure (B = 1000 repetitions).
Tumor Immunoassay
To explore the relationship between the risk models and immune infiltration status, immune-related files were downloaded. The immune infiltrating cell file download address (http://timer. comp-genomics.org) was accessed on June 18, 2022. Immune escape file download address (http://tide.dfci.harvard.edu/) was accessed on June 23, 2022, using “limma” and “pheatmap” packages to analyze the differences between the high- and low-risk groups.
Tumor mutational burden analysis
Mutation data for NSCLC after body radiotherapy were downloaded from the TCGA database, and the Perl programming language (version strawberry-Perl-5.30.0; https://www.perl.org) was used to obtain the mutation data. The 50 genes with the highest mutation frequencies were visualized using the “maftools” package. The tumor mutational burden was subjected to survival analysis using the ”maftools”package.
Cell culture
The NSCLC cell lines A549 and H1299 were cultured in Dulbecco’s modified DMEM medium (DMEM; Gibco) containing 10% fetal bovine serum (FBS; Gemini) and 10% streptomycin-penicillin (Gibco) at 37°C and 5% CO2.
Quantitative real-time PCR
Total RNA was extracted from radiotherapy-treated A549 and H1299 cells using the TRIzol reagent (Invitrogen). RNA was converted into cDNA using HiScript III RT SuperMix for qpCR (+gDNA wiper) (Vazyme Biotech Co., Ltd.). Real-time PCR was performed using SuperReal PreMix Plus (SYBR Green) (TIANGEN), and the Thermo Fisher Scientific Applied Biosystems QuantStudio5 real-time PCR system was used to detect the expression level of lncRNA-LINC02029. The 2-ΔΔCT method was used to calculate the relative gene levels. The primer sequences are listed in table 1.
Table 1.
Primer sequences for qRT-PCR.
Primer |
Sequence |
LINC02029-F |
TAGAGATGGAGGACTGGGAGG |
LINC02029-R |
GTGCACACTTGTCCAAGCAG |
GAPDH-F |
GTCTCCTCTGACTTCAACAGCG |
GAPDH-R |
ACCACCCTGTTGCTGTAGCCAA |
Statistical Analysis
All statistical analyses were performed using the R software (version 4.1.3). Differential mRNAs and lncRNAs associated with cuproptosis and risk differential genes were identified using the "limma" package. Univariate and multivariate COX analyses were used to construct the prognostic models. The "Survival" and "survminer" packages were used to analyze the relationship between cuproptosis-related lncRNAs and overall survival. The "Survival" package was used for univariate and multivariate independent prognostic analyses. The "survivalROC" package was used to draw the ROC curves and determine the area under the curve (AUC) values.
- Under my point of view, an extensive revision of English should be performed because I have found several grammatical mistakes in all sections of the text. For example several phrases were difficult to understand (Example in Material and Methods: “We downloaded normalized RNA-seq data for radiotherapy-treated NSCLC from the TCGA website (https://portal.gdc.cancer.gov/), along with corresponding clinical and mutational data; 180 primary tumors and were obtained Example solid tissue normal data. Clinical data of 178 patients who underwent radiation therapy. (http://timer.cistrome.org) website to download information on immune infiltration. (http://tide.dfci.harvard.edu/) website to download data on immune escape”).
We thanked the insightful question of the reviewer. About this question, The full article has been extensively revisioned, and thanks to editor Eleanor H for polishing the full article.
- Authors should include more details in figure legends since descriptions of figures are very simple, and in some cases difficult to understand.
We thanked the insightful question of the reviewer. About this question,Additions were made as below.
Figure 2. (A) The Y-axis of the Sankey diagram represents cuproptosis-related genes, while the X-axis represents the related lncRNA. (B) The lncRNAs involved in the model construction. (C) The abscissa represents the lncRNAs involved in the model construction, and the ordinate represents the cuproptosis-related gene. Blue indicates a negative correlation, while red indicates a positive correlation. (D) Gene and LncRNA co-expression network. (E) The expression pattern of LINC02029 prognostic lncRNAs in H1299 and A549 cells. ***p<0.001,****p<0.0001
Figure 3. Kaplan–Meier survival analyses of patients
OS rates of NSCLC patients in high-risk and low-risk groups (A、D、G).The number of dead and alive patients with different risk scores in different groups. Blue represents the number of survivors and red represents the number of deaths (B, E, H). Risk curves represent the distributed survival status of NSCLC patients with different risk scores (C, F, I). A, B, C, all groups; D, E, F, the train group; and G, H, I, the test group.
Figure 4. Kaplan–Meier survival analyses of patients
- PFS of high and low risk groups. (B, C) Stage classification of clinicopathological variables. (D, E, F) LncRNA risk heat map of 177 samples, Train group and Test group.
Figure 5. (A) Forest plot of univariate Cox regression analysis. (B) Forest plot of multivariate Cox regression analysis. (C) AUCs for 1-, 3- and 5-year survival in the predictive signature AUC, area under the curve. (D) C-index curve analyzed the concordance index of the risk score.
Figure 6. (A) Nomograms of clinicopathological variables and risk scores to predict 1-, 3-, and 5-year OS in patients with NSCLC. (B) The Calibration curves for the nomogram. The x-axis represents the nomogram-predicted OS probability, and the y-axis represents the Observed OS probability of NSCLC patients. Perfect prediction would correspond to the 45°gray line. The green, blue, and red lines represents the OS at 1, 3, and 5 years, respectively, estimated by bootstrapping (B=1000 repetitions), indicating the observed nomogram performance.
Figure 7. Results of principal component analysis. PCA analysis of the distribution of patients according to A. Risk lncRNA, B. All genes, C. Genes associated with cuproptosis, and D. lncRNAs associated with cuproptosis.
Figure 8. (A) CD276, Recombinant Cluster Of Differentiation 276[23]; TNFSF14, Recombinant Tumor Necrosis Factor Ligand Superfamily, Member 14[24]. (B) TIDE, Tumor Immune Dysfunction and Exclusion, *p<0.05. (C) Methods for different immune prediction software
4) The “Conclusions” sections should be renamed as “Discussion”. In this respect, authors have referenced others studies in this section to discuss their findings. I think that they should included 1 or 2 paragraphs to discuss their own findings because of the novelty of their study.
We thanked the insightful question of the reviewer. About this question,Additions were made in the discussion。
LncRNAs play an important regulatory role in NSCLC radiotherapy. lncRNA CRNDE/PRC2 targeting p21 enhances radioresistance in NSCLC [40]; SPRY3-2/3/4 and H19–miR-130a-3p–WNK3 axis modulates radiosensitivity and affects apoptosis and cell viability in NSCLC [41][42]. However, lncRNAs associated with radiotherapy-related cuproptosis have never been studied in NSCLC.In our Studies have shown that LUCAT1 is significantly upregulated in NSCLC, where it represses the expression of p21 and p57 by associating with PRC2 to mediate epigenetic regulation. Development of cisplatin resistance in NSCLC is closely related to autophagy, affects tumor proliferation, and is closely linked to patient prognosis[43][44][45]. HHLA3 is an important member of the clear cell renal cell carcinoma cuproptosis-related prognostic model [46] and is closely related to the metabolic immune infiltration of NSCLC [47]. The remaining lncRNAs described here were reported for the first time. In particular, these newly discovered NSCLC-related lncRNAs may serve as new targets for cancer therapy.
Minor issues
1) Statistical analyses performed should be showed in all figure legends.
We thanked the insightful question of the reviewer. About this question,Additions were made as question 3.

Round 2
Reviewer 1 Report
The authors made significant changes to the manuscript and answered most of my concerns. Although adding supplementary data, as described in the text, is much appreciated, I could not access the files provided with the submission.
Minor comments upon reading the revised version of the manuscript:
-The y-axis title in Figure 2E is missing.
-Line 404: In our Studies have shown, edit for clarity
Author Response
Dear reviewer
First of all, thank you for your insightful comments on the article, which have provided us with further opportunity to improve. Secondly, thank you for your affirmation of our efforts. I apologize for not sending you a supplementary data the first time. Because the interface we can only upload one word /PDF and cannot upload the compressed package, so we upload the compressed package to the home page.
Sincere thanks
Minor comments upon reading the revised version of the manuscript:
We thanked the insightful question of the reviewer. About this question, which has been modified accordingly as follows:
-The y-axis title in Figure 2E is missing.
-Line 404: In our Studies have shown, edit for clarity
We thanked the insightful question of the reviewer. About this question, the corresponding additions have been made as follows
Therefore, In our Studies have shown,the identification of lncRNAs associated with cuproptosis in patients with NSCLC undergoing radiotherapy is of great significance for early diagnosis, combination therapy, prognostic analysis, and reduction of radiation resistance.

Reviewer 2 Report
Dear authors,
Congratulations again for your performed word. The slight issues were addressed correspondingly.
Kind regards,
Author Response
Dear reviewer
First of all, thank you for your insightful comments on the article, which have provided us with further opportunity to improve. Secondly, thank you for your affirmation of our efforts. We wish you all the best.
sincere regards
Reviewer 3 Report
Authors should better explain panels of figure Legends and include the statistical analyses performed en each figure Legend.
Author Response
Dear reviewer
First of all, thank you for your insightful comments on the article, which have provided us with further opportunity to improve. Secondly, thank you for your affirmation of our efforts. Hopefully, our revisions have largely aligned with your valuable comments.
Sincere thanks
Authors should better explain panels of figure Legends and include the statistical analyses performed en each figure Legend.
We thanked the insightful question of the reviewer. About this question, We have made extensive changes to the figure legend,the corresponding additions have been made as follows
Figure 2. (A) Sankey diagram for the co-expression network in NSCLC. The above axis of the Sankey diagram represents cuproptosis-related genes, while the below axis represents the related lncRNA. (B) Harzard ratio (95% CI) of differnet lncRNA by cox regression.Red represents high-risk HR>1; green represents low-risk HR <1(C) Heatmap plots of the differentially expressed lncRNAs. The horizontal axis represents risk LncRNA,The vertical axis represents cuproptosis-related genes. The color represents the correlation of the two. Blue indicates a negative correlation, while red indicates a positive correlation. (D) Cytoscape software builds the Gene and LncRNA co-expression network, yellow for risk prediction LncRNA, green for co-expression of cuproptosis genes. (E) Real-time Quantitative PCR analysis of the expression pattern of LINC02029 prognostic lncRNAs in H1299 and A549 cells. *p<0.05,**p<0.01,***p<0.001,****p<0.0001
Figure 3. Kaplan-Meier curve evaluation (ADG)The overall survival rates of NSCLC patients in various categories.Abscissa: Time; Ordinate: survival rate;Each point represents the patient's survival rate at that moment.(B, E, and H) The number of survivors and deaths at various risk levels.The colors blue and red symbolise survival and death, respectively.Survival status curves of patients with varying risk ratings (C, F, I).A, B, C represent all groups; D, E, F represent the train group; and G, H, I represent the test group.
Figure 4. The Kaplan-Meier curve for progression-free survival in the high and low risk groups, with the abscissa representing time and the ordinate indicating progression-free survival.(B, C) Validation of clinical grouping models in stages I-II and III-IV.(D, E, F) LncRNA risk heat map of 177 samples, with the abscissa representing the high and low risk groups and the ordinate representing the risk prediction LncRNA.
Figure 5. (A) Harzard ratio (95% CI) forest plot of a univariate Cox regression analysis.(B) Forest plot of multivariate Cox regression analysis using independent prognostic variables and the Harzard ratio (95% CI).(C) The predictive signature's AUCs for survival.The ordinate is the genuine positive rate annotation method, while the abscissa is the false positive rate annotation method with 1-specificity.Sensitivity is represented by the red, yellow, and green ROC curves, with red denoting one year, yellow three years, and green five years.(D) The concordance index of the risk score was evaluated utilizing C-index curve.The red curve represents the risk score, the yellow curve represents age, the green curve represents gender, and the blue curve represents stage. Abscissa represents time, ordinate represents C-index value, estimated by bootstrapping (B=1000 repetitions), and the larger the value, the higher the accuracy in the model's prediction.
Figure 6. (A) Clinicopathological variable nomograms and risk scores to predict 1-, 3-, and 5-year OS in NSCLC patients. (B) The nomogram calibration curves. The x-axis indicates the nomogram-predicted OS probability, while the y-axis represents the NSCLC patients' observed OS probability. The 45°gray line would equate to a perfect forecast. The green, blue, and red lines reflect the observed nomogram performance at 1, 3, and 5 years, respectively, as assessed by bootstrapping (B=1000 repetitions
Figure 7. The results of principal component analysis.PCA analysis of patient distribution according to A. Risk lncRNA, B. All genes, C. Cuproptosis-associated genes, and D. Cuproptosis-associated lncRNAs.
Figure 8. (A) Immune checkpoint analysis in high- and low-risk groups, in which the ordinate represents the level of gene expression and the abscissa the immune checkpoint-related gene. TNFSF14, recombinant Tumor Necrosis Factor Ligand Superfamily, Member 14[24]; CD276, recombinant Cluster Of Differentiation 276[23]. (B) In the TIDE (Tumor Immune Dysfunction and Exclusion) abscissa, the high and low risk categories are represented by blue and red, respectively. TIDE score *p<0.05 serves as the ordinate. (C) Relationship analysis of immune cells in high- and low-risk groups. The ordinate represents immune cells, the abscissa represents high- and low-risk groups, and the various colors represent the predictions from various software programs.
Figure 9. (A, B) A waterfall plot depicts the top 50 genes in the high and low risk categories, respectively. Each abscissa represents a single sample, the left ordinate represents the gene, the right ordinate shows the frequency of gene mutation, and the different colors of the legend reflect different mutation types. (C, D) Tumor mutation burden survival curve, abscissa represents survival time, ordinate represents survival rate, red represents high mutation burden group, blue represents low mutation burden group.
